# Comparative Analysis of Skew-Join Strategies for Large-Scale Datasets with MapReduce and Spark

Anh-Cang Phan [1,*] , Thuong-Cang Phan [2,*] , Hung-Phi Cao [1] and Thanh-Ngoan Trieu [2,3]

1   Faculty of Information Technology, Vinh Long University of Technology Education,
    Vinh Long 85110, Vietnam; caohungphi@vlute.edu.vn
2   College of Information and Communication Technology, Can Tho University, Can Tho 94115, Vietnam;
    ngoan.trieuthanh@etudiant.univ-brest.fr
3   La Faculté Sciences et Techniques, Université de Bretagne Occidentale, 29200 Brest, France
*   Correspondence: cangpa@vlute.edu.vn (A.-C.P.); ptcang@cit.ctu.edu.vn (T.-C.P.);
    Tel.: +84-918-204-917 (A.-C.P.); +84-292-3831-301 (T.-C.P.)

**Abstract:** In the era of data deluge, Big Data gradually offers numerous opportunities, but also poses significant challenges to conventional data processing and analysis methods. MapReduce has become a prominent parallel and distributed programming model for efficiently handling such massive datasets. One of the most elementary and extensive operations in MapReduce is the join operation. These joins have become ever more complex and expensive in the context of skewed data, in which some common join keys appear with a greater frequency than others. Some of the reduction tasks processing these join keys will finish later than others; thus, the benefits of parallel computation become meaningless. Some studies on the problem of skew joins have been conducted, but an adequate and systematic comparison in the Spark environment has not been presented. They have only provided experimental tests, so there is still a shortage of representations of mathematical models on which skew-join algorithms can be compared. This study is, therefore, designed to provide the theoretical and practical basics for evaluating skew-join strategies for large-scale datasets with MapReduce and Spark—both analytically with cost models and practically with experiments. The objectives of the study are, first, to present the implementation of prominent skew-join algorithms in Spark, second, to evaluate the algorithms by using cost models and experiments, and third, to show the advantages and disadvantages of each one and to recommend strategies for the better use of skew joins in Spark.

**Keywords:** big data analytics; skew join; MapReduce; Apache Spark

## 1. Introduction

Big Data is a term that has been mentioned in many recent studies. People generate terabytes of data every hour, leading to challenges in storing and handling data in traditional ways. Therefore, Google designed the MapReduce processing model [1] for parallel and distributed processing of large-scale datasets. One of the major limitations of the MapReduce model is data-skew processing [2]. A typical case is data skew in join operations, which are the operation of joining many relationships or datasets that have some common attributes into a new relationship [3]. The join operation is used in many applications, such as the construction of search engines and some big data-intensive applications [4]. Taking the example of joining two datasets $R$ and $L$, $R$ has the join key column $R.id = \{1,1,1,1,1,1,2\}$ and $L$ the has join key column $L.id = \{1,2,3\}$. In MapReduce, Mappers read input data from the two datasets and distribute intermediate data to reducers based on join keys. Data with the same join keys will be sent to the same reducers to create join results. Assuming that dataset $R$ is skewed at $id = 1$, the join key with a value of 1 occurs many times. In this case, the reducers $r_2$ and $r_3$ will finish faster than reducer $r_1$, since reducer $r_1$ receives more data than the others. If there is a large amount of skewed data with $id = 1$, then

overload and congestion will appear at the computing node $r_1$ and lead to inefficiency in the computational system. This is a real challenge for join operations in MapReduce.

Some algorithms have been proposed for data-skew-join operations in MapReduce, such as hash-based partition [5], range-based partition [5], multi-dimensional range partition (MDRP) [6], MRFA-Join [7], and randomized partition [8]. Chen et al. [9] introduced LIBRA, a sampling and partitioning algorithm for handling high-frequency join keys in reduction tasks. Bruno et al. [10] discussed challenges in large-scale distributed join operation and introduced novel execution strategies that robustly handle data skew. They used different partitioning schemes and join graph topologies for high-frequency key values. Zhou et al. [11] proposed an efficient key-select algorithm to find skew key tuples and defined a lightweight tuple migration strategy to solve data-skew problems. Their FastJoin system improved the performance in terms of latency and throughput. Zhang and Ross [12] presented an index structure to reorder data so that popular items were concentrated in the cache hierarchy. They analyzed the cache behavior and efficiently processed database queries in the presence of skew. Meena et al. [13] presented their approach for handling data skew in a character-based string-similarity join in MapReduce. They compared the proposed algorithm with three other algorithms for handling data skew for evaluation. Jenifer and Bharathi [14] gave a brief survey on the solutions for data skew in MapReduce. Nawale and Deshpande [15] studied various methodologies and techniques used to mitigate data skew and partition skew. Myung et al. [6] proposed multi-dimensional range partitioning to overcome the limitations of traditional algorithms. There have been some studies on data skew in MapReduce. However, there has not been any study that has shown an adequate and systematic comparison of data-skew handling for join operations in Spark.

In this paper, we will present and compare three algorithms for partitioning data, i.e, hash-based partition, range-based partition, and multi-dimensional range partition. This helps users to choose suitable solutions for processing join operations in large-scale datasets. We built cost models for the algorithms for evaluation. This provided a scientific basis for the comparison. Experiments were conducted on a Spark cluster with different skew ratios. The structure of this paper is organized as follows. Section 2 presents the background related to the large-scale data processing model and platform. Section 3 provides the solutions for data skew in join operations on large datasets in detail. Section 4 presents an evaluation with the cost models and experiments conducted in the Spark cluster. The conclusion of the paper is presented in Section 5.

## 2. Background

### 2.1. Join Operations in the MapReduce Model

MapReduce [1] is a parallel and distributed large-scale data processing model. A program can be run on clusters with a number of computing nodes that can be up to thousands. Introduced in 2004, MapReduce has been widely used in the field of Big Data, since it allows users to simply focus on the design of data processing operations regardless of the parallel or distributed nature of the model [3]. MapReduce is implemented through two basic functions, Map and Reduce, which are also two consecutive stages in data processing. The Map function receives input data to convert them into intermediate data (key–value pairs), and the Reduce function accepts the intermediate data created to perform calculations.

Hadoop (http://hadoop.apache.org (accessed on 8 May 2022)) has become one of the popular Big Data processing platforms of the last decade [16]. Hadoop is an open-source implementation of the MapReduce model. Hadoop breaks data down into many small chunks and runs an application on the data of the computing nodes in the system. Each time it performs a task, Hadoop has to reload the data from the disk, which is costly and is considered a "penalty" [17]. Therefore, Hadoop has not fully supported join operations with high I/O and communication costs [2]. In recent years, with the advent of Apache Spark [17], many outstanding features of this platform have helped this to become the next generation of Big Data processing platforms [6].

The join operation is a basic operation that consumes much processing time, and it is an intensive data operation in data processing [3]. This section discusses the process and gives a concrete example of a join operation using the MapReduce model. Considering the two datasets $R$ (user data) and $L$ (log data), to list a username and the corresponding event that the user accesses, we have the query $R(uname, uid) \bowtie_{uid=uid} L(uid, event)$. In large-scale data applications, such as social networks, this query can enforce the joining of trillions of records. Therefore, the parallel and distributed processing model in MapReduce is a good solution to this problem. As shown in Figure 1, the JobTracker of MapReduce creates three Mappers to process three partitions of the input data simultaneously. The first Mapper computes the first partition, consisting of three records of dataset $R$. The second Mapper processes one record of $R$, and the third Mapper processes two records of $L$. The Mappers transform the datasets $R$ and $L$ based on the join key *uid*.

- $\{(A, B), (C, B), (A, F)\} \rightarrow \{(B, A), (B, C), (F, A)\}$
- $\{(C, D)\} \rightarrow \{(D, C)\}$
- $\{(B, C), (D, F)\} \rightarrow \{(B, C), (D, F)\}$

The data after transformation are called intermediate data, and those are sent to the Reducers. Intermediate datasets with the same join key are sent to the same Reducer. Here, the reduction function is called for every single key with a list of values. Finally, each record of $R$ finds the records of $L$ that have the same join key to produce the join results. Figure 1 shows that there are three join results from the Reducers.

- Reducer 1 for join key B: $(B, [R : A, R : C, L : C]) \rightarrow \{(A, C), (C, C)\}$
- Reducer 2 for join key F: $(F, [R : A]) \rightarrow \{empty\}$
- Reducer 3 for key join D: $(D, [R : C, L : F]) \rightarrow \{(C, F)\}$

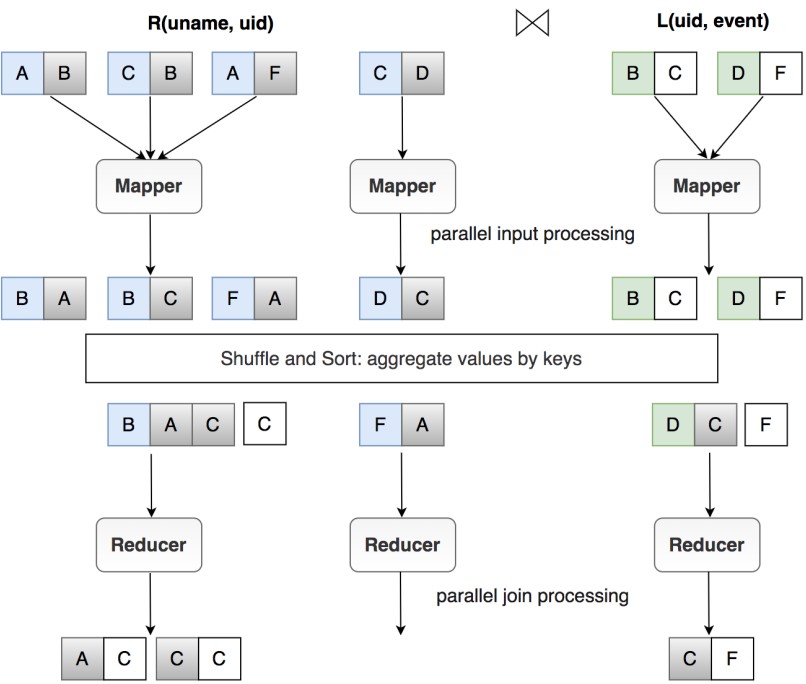

**Figure 1.** Join operation in MapReduce.

### 2.2. Apache Spark

Apache Spark (http://spark.apache.org (accessed on 8 May 2022)) is an open-source cluster computing framework that was originally developed in 2009 by AMPLab at the University of California, Berkeley. Spark has continued to be developed by the Apache Software Foundation since 2013. Given a task that is too large to be handled on a server, Spark allows us to divide this task into more manageable tasks. Then, Spark will run

these small tasks in memory on a cluster of computing nodes. Apache Spark has three salient features.

- Speed: Spark is 100 times faster when running in memory and 10 times faster when running on a disk than Hadoop [17].
- Support for multiple programming languages: Spark provides built-in APIs in the Java, Scala, and Python languages.
- Advanced analysis: Spark not only supports MapReduce models, but also supports SQL queries, streaming data, machine learning, and graph algorithms.

### 2.2.1. Resilient Distributed Datasets

Resilient distributed datasets (RDDs) are the underlying data structure and outstanding feature of Spark. They are a type of distributed collection that can be temporarily stored in RAM with a high fault tolerance and the capability of parallel computation. Each RDD is divided into multiple logical partitions and can be computed on different nodes of a cluster. In this research, we use the Scala language, since Apache Spark is built mainly on Scala, so it has the best speed and support for this language. RDDs basically support two main types of operations.

- Transformation: Through a transformation, a new RDD is created from an existing RDD. All transformations are "lazy" operations, meaning that these transformation operations will not be performed immediately, but the steps taken are only memorized and saved as pending scripts. This process can be understood as a job-planning process. Those operations can only be performed when an Action is called.
- Action: An Action performs all transformations related to it. By default, each RDD will be recalculated if an Action calls it. However, RDDs can also be cached in RAM or on a disk using the persist command for later use. The Action will return the results to the driver after performing a series of computations on the RDDs.

It would be time-consuming if we encountered an RDD being reused many times because each RDD will be recalculated by default. Therefore, Spark supports a mechanism called persist or cache. When we ask Spark to persist with an RDD, the nodes that contain those RDDs will store those RDDs in memory, and that node will only compute once. If the persist fails, Spark will recalculate the missing parts if necessary.

### 2.2.2. Examples of Spark Functions

Map (a transformation) returns a new RDD by passing each input element through a function. An example of a map function is presented in Figure 2, in which each element in $x$ is a map with 1 and creates a new RDD $y$.

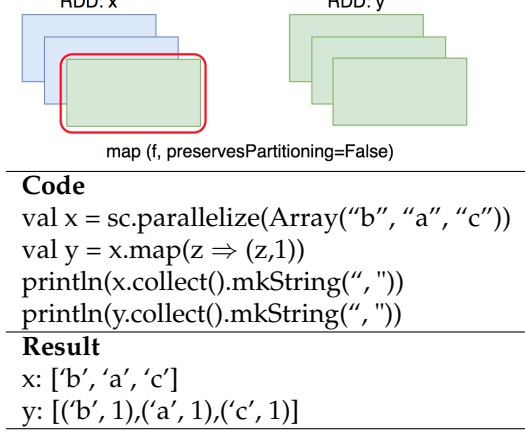

| Code |
| --- |
| val x = sc.parallelize(Array("b", "a", "c")) |
| val y = x.map(z ⇒ (z,1)) |
| println(x.collect().mkString(", ")) |
| println(y.collect().mkString(", ")) |
| **Result** |
| x: ['b', 'a', 'c'] |
| y: [('b', 1),('a', 1),('c', 1)] |

**Figure 2.** Map function.

Reduction (an action) aggregates all elements of the original RDD by applying a user function in pairs with the elements and returns the results to the driver. An example of a reduction function is presented in Figure 3, in which the RDD *x* is reduced to the sum of all its elements.

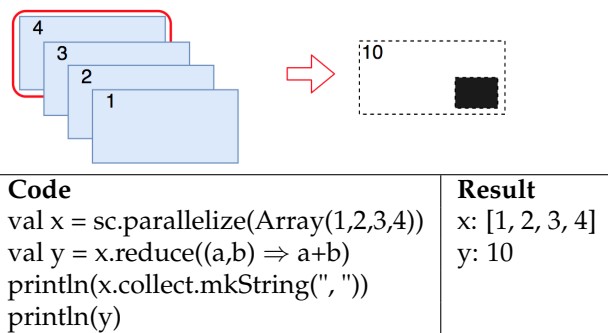

| Code | Result |
|------|--------|
| val x = sc.parallelize(Array(1,2,3,4)) | x: [1, 2, 3, 4] |
| val y = x.reduce((a,b) ⇒ a+b) | y: 10 |
| println(x.collect.mkString(", ")) | |
| println(y) | |

**Figure 3.** Reduction function.

## 3. Data-Skew-Handling Algorithms

### 3.1. Skew Join

Data skew is a problem in which the data distribution is uneven or asymmetric. In a database, cases where some attributes appear with a greater frequency than other attributes usually occur [18], and this is called data skew. In distributed parallel computation systems, the join operations using these attributes will take a longer time than with data with a normal distribution. A join operation consists of several steps, including data uploading, projection and selection based on the query, partitioning of the data into partitions, and joining of datasets. Therefore, data skew limits the efficiency of join operations in parallel computation.

There are several types of data skew based on the stage in which skew problems occur, such as tuple placement skew, selectivity skew, redistribution skew, and join skew [19]. The initial placement of the datasets in the partitions may cause a tuple placement skew. Selectivity skew occurs when the selectivity of selection predicates is different between nodes. Redistribution skew is caused by a large amount of data on the partitions after the redistribution scheme is applied. Joining product skew is the result of join selectivity on each node. In this paper, we will not consider tuple placement skew, since MapReduce creates split files regardless of the size of the original file. Selectivity skew is also ignored because it does not have any significant impact on the performance, and we do not use projections and selections in our program. Joining product skew cannot be avoided, since it is created after joining two datasets together. Redistribution skew is the main and most important type of data skew, and it affects the distribution of the workload between nodes. This situation is caused by an improper redistribution mechanism. Hence, we will cover redistribution skew in our research.

Join computations based on the MapReduce model go through two phases, Map and Reduce. Mappers read the data and convert them into intermediate data in the form of key–value pairs, with the key being the join key and the value being the row containing the join key. After generating key–value pairs, Mappers will shuffle the intermediate data to Reducers according to the rule that pairs of the same key will go to the same Reducers. Data skew may appear at this stage if the input data contain one or more frequent join keys (Figure 4a). This will inevitably lead to data imbalances between computing nodes. Data containing these frequent join keys are processed by only one or a handful of computing nodes, while the rest of the computing nodes are idle. Thus, skewed data occur when one or more computing nodes have to process a much larger number of join keys than other computing nodes in a system [20]. As a result, some nodes encounter bottlenecks or delays, and the remaining nodes are a waste of resources. In parallel computing, the join operation

execution time is determined by the longest-running task. Therefore, if the data are skewed, the benefits of parallel computation become meaningless [2].

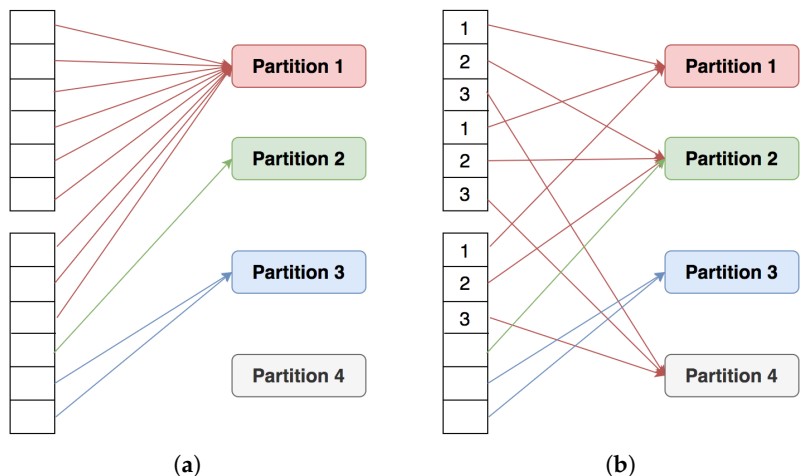

(**a**)           (**b**)

**Figure 4.** Data partitioning example. (**a**) Skew partitioning. (**b**) Not skewed.

A solution is to clearly point out skew data and to allow it to be distributed in different ways to avoid or reduce the skew effect before calculations begin. In a data-skew problem, heavily weighted partitions will appear. Spark assigns one task per partition, and each worker processes one task at a time; thus, heavy partitions will affect the performance. The main idea now is to avoid heavily weighted partitions. As an example, we add more information on the skew-join key so that it can be distributed into different partitions (Figure 4b).

In Spark, data are divided into partitions on many different nodes in a cluster. Thus, it is difficult to avoid data shuffling between nodes with join operations in Spark. This shuffling process will slow down the processing speed and program performance. Therefore, reasonable partitioning of data before the join computation can improve the performance and reduce the effect of shuffling of data. Figure 5 presents the data flow of the three algorithms.

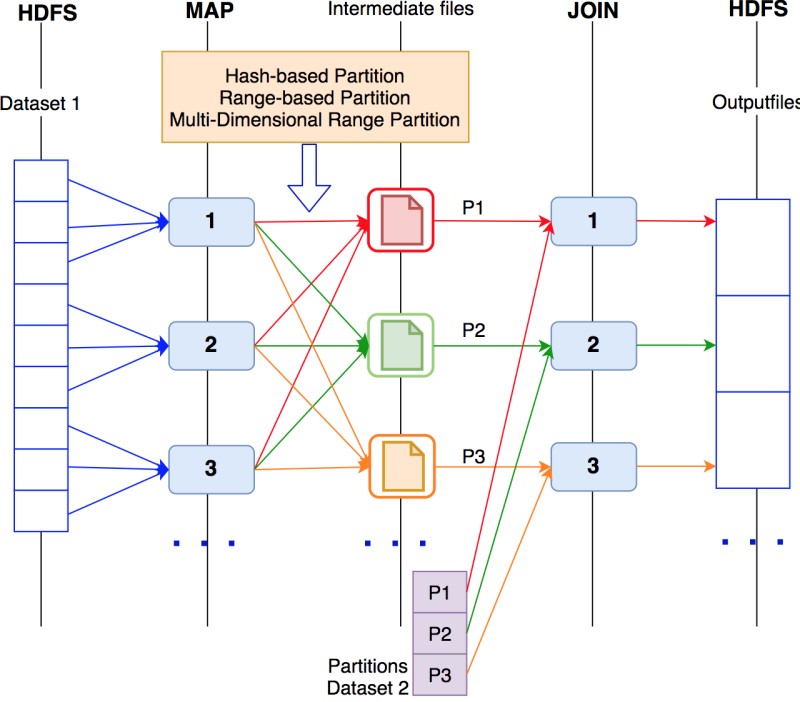

**Figure 5.** Data flow of the three join algorithms.

*3.2. Hash-Based Partition*

In hash-based partition (HBP) [5], after mapping the input data, the partition processing will redistribute the workload to the nodes based on the join keys. Hash-based partition is not ideal for handling skewed data, since the data skew will go to the same computing nodes. Therefore, hash-based partition is not a good solution for skewed data [21].

Suppose that we have two datasets $R(k, v1)$ and $L(k, v2)$, where the attribute $k$ is a join key. The data flow of hash-based partition algorithm is presented in Figure 6. The join operation between two datasets $R$ and $L$ using a hash-based partition algorithm goes through two phases.

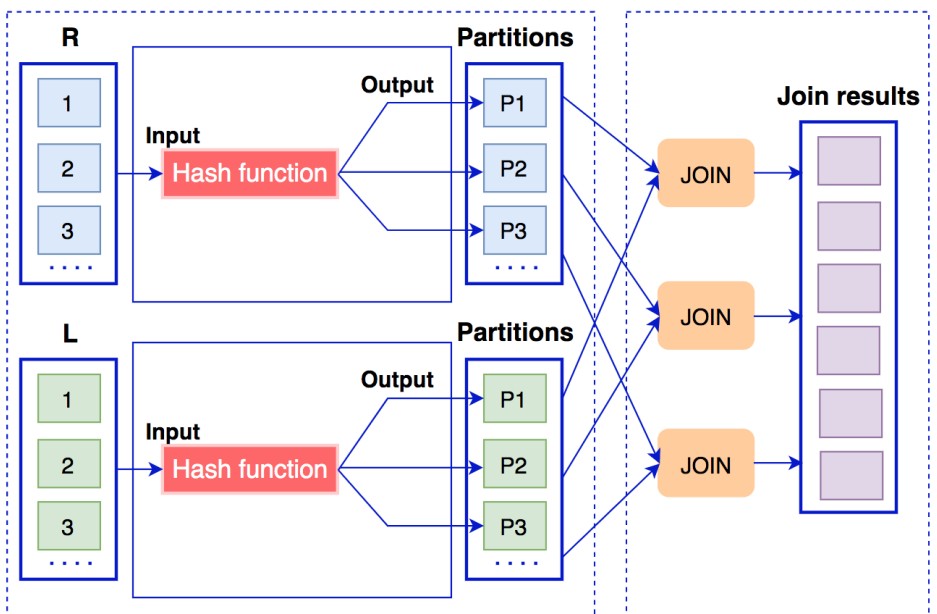

**Figure 6.** Data flow of a hash-based partition algorithm.

- Partitioning phase: The $k$ attribute in each record of the dataset $R$ is passed to a hash function by the formula $(R.k \bmod p)$, where $p$ is the number of partitions. The result of the hash function is also the partition number to which this record was sent. The same thing is done with dataset $L$.
- Join phase: The partitions will receive a list of data with the same join key $k$. Here, we can use any join algorithms, such as join, rightOuterJoin, or leftOuterJoin.

Given an example of two datasets $R(k, v1)$ and $L(k, v2)$ with $R.k = \{1, 1, 2, 2, 2, 2, 4, 4\}$, $L.k = \{1, 1, 2, 2, 2, 3, 4, 4\}$, and there are $p = 4$ partitions, in hash-based partition, one partition can receive more records than other partitions. The join computation with hash-based partition is shown in Figure 7. In this example, partition $P_2$ has four records $\{r3, r4, r5, r6\}$ from dataset $R$ and three records $\{l3, l4, l5\}$ from dataset $L$. Thus, partition $P_2$ produces 12 join results $\{(r3, l3), (r3, l4), (r3, l5), (r4, l3), (r4, l4), (r4, l5), (r5, l3), (r5, l4), (r5, l5), (r6, l3), (r6, l4), (r6, l5)\}$, which is a larger number than in other partitions. Specifically, partition $P_3$ only receives one record $\{l6\}$ and does not generate any join results. The execution time of a join computation depends on the completion of the last reducer. If the number of identical join keys is too large, then, even if the number of partitions is large, the data with the same join key will only gather on a few certain partitions. This results in some partitions being too big and the others having no data. This easily causes out-of-memory errors or slows down the processing speed. Therefore, processing skewed data is a very important issue for join operations.

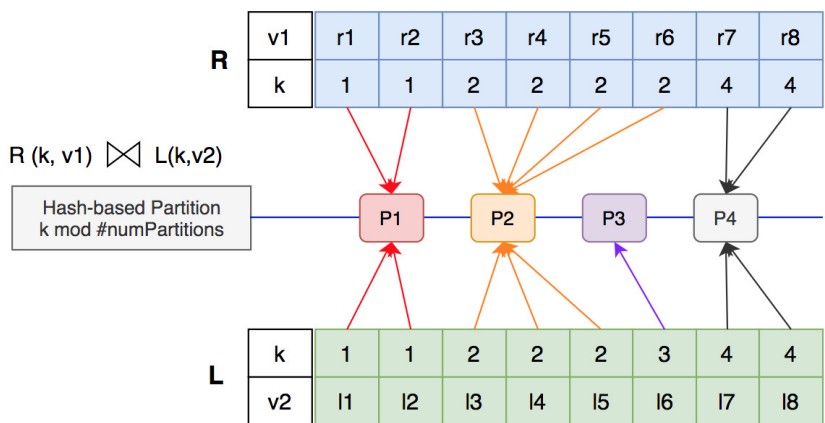

**Figure 7.** Join computation with hash-based partition.

*3.3. Range-Based Partition*

Range-based partition (RBP) is proposed as a solution for skewed data instead of hash-based partition [5]. The idea of the algorithm is to divide the mapped data into sub-ranges. Therefore, each computing node will process a sub-range instead of a join key value in order to reduce the burden on the computing nodes when data are skewed. The data flow of range-based partition algorithm is presented in Figure 8. A split vector is created to limit the range of sub-ranges. If there are $n$ partitions, the split vector will contain $n-1$ elements $\{e_1, e_2, e_3, \ldots, e_{n-1}\}$. Thus, data with join keys $\leq e_1$ will come to partition 1, data with join keys in the range $e_1 < keys \leq e_2$ will come to partition 2, and data with join keys $> e_{n-1}$ will come to partition $n$. However, the algorithm still produces skewed results after the partition period.

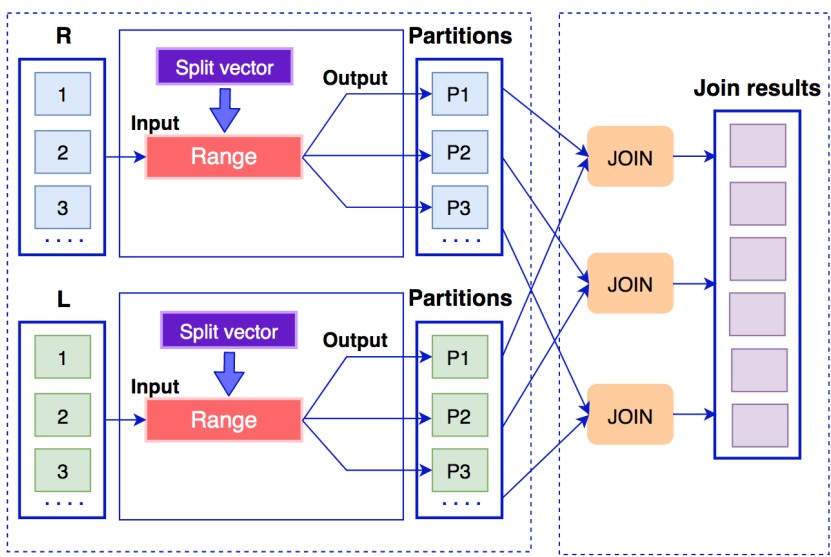

**Figure 8.** Data flow of a range-based partition algorithm.

In this method, the first thing to do is to create a split vector from the two input datasets by calculating the appearance of the join keys in each dataset. The dataset that has the most skewed join keys will be selected as the split vector. We use the "fragment–replicate" technique [22] before the join computation. When a mapper reads input records and maps into RDD key–value pairs, if the join key belongs to more than one partition, this key–value pair is a fragment or replicated. We will use a fragment of the key–value pairs on the dataset with the most skewed join key and use a replicate of the key–value pairs on the other dataset.

The following steps need to be performed to partition key–value pairs in the dataset with more skewed join keys (a build relation):

- A split vector is created from the input dataset and this vector is stored in the HDFS;
- Each node in the cluster will read the split vector from the HDFS and create a range map, and each sub-range will be assigned to the corresponding partition;
- Based on this range map, when a mapper reads input records and maps into RDD key–value pairs, if the join key belongs to only one sub-range, the data will be partitioned into the corresponding partition; if the join key belongs to more than one sub-range, the data will be randomly assigned to one of the partitions corresponding to the sub-ranges (fragment).

For a dataset with fewer skewed join keys, we use the split vector created above to partition key–value pairs (a probing relation). If the join key belongs to only one sub-range, the data will be assigned to the corresponding partition. Conversely, if the join key belongs to more than one sub-range, the data will be distributed to all partitions corresponding to the sub-range (replicate). The algorithm used to partition the datasets is presented in Algorithms 1 and 2.

---

**Algorithm 1** Range-based partition algorithm for dataset $R$—a build relation

---

**Input:** An input record $r \in R$
**Input:** Split vector
**Input:** A list of partitions $P$
**Begin**

```
 1: Create Sub-range
 2: for each r ∈ R do
 3:     SR = r.listSubRange()
 4:     if |SR| == 1 then
 5:         Output (P(SR(1)), r)
 6:     else
 7:         i = random() % |SR|
 8:         Output (P(SR(i)), r)
 9:     end if
10: end for
```

**End.**

---

**Algorithm 2** Range-based partition algorithm for dataset $L$—a probing relation

---

**Input:** An input record $l \in L$
**Input:** Split vector
**Input:** A list of partitions $P$
**Begin**

```
 1: Create Sub-range
 2: for each l ∈ L do
 3:     SR = l.listSubRange()
 4:     if |SR| == 1 then
 5:         Output (P(SR(1)), l)
 6:     else
 7:         for i = 1 to | SR | do
 8:             Output (P(SR(i)), l)
 9:         end for
10:     end if
11: end for
```

**End.**

---

The partitions will have a list of data with a join key $k$ belonging to the same sub-range. Here, we use a join algorithm to create the join results (presented in Algorithm 3).

---

**Algorithm 3** Join two datasets $R \bowtie L$

---

**Input:** (partitionID, $\{r1, r2, \ldots, l1, l2, \ldots\}$)
**Begin**

1: tupleR = {}
2: **for each** $r_i$ in input list **do**
3:     tupleR = tupleR $\cup \{r_i\}$
4: **end for**
5: **for each** $l_j$ in input list **do**
6:     **for each** $r_i$ in tupleR **do**
7:         Output $r_i \bowtie l_j$
8:     **end for**
9: **end for**
**End.**

---

Given an example of two datasets $R(k, v1)$ and $L(k, v2)$, as presented above, we create a split vector from dataset $R$. The elements in the split vector will be chosen from $R.k$ with the formula $\lfloor \frac{|R|}{p-1} \rfloor$, in which $|R|$ is the number of records of dataset $R$. In the example (Figure 9), we have $|R| = 8$ and $p = 4$; thus, the 2nd, 4th, and 6th elements in $R.k$ will be chosen to have the split vector $\{1, 2, 2\}$. From the split vector, we create four sub-ranges, i.e., $[-\infty, 1], (1, 2], [2, 2], (2, \infty]$.

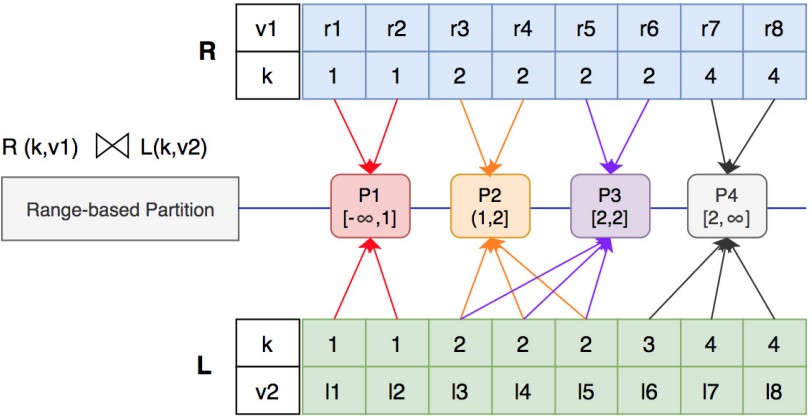

**Figure 9.** Join computation with range-based partition.

According to the "fragment–replicate" technique, $\{r3, r4, r5, r6\}$ are classified into two partitions and $\{l3, l4, l5\}$ are replicated in two partitions ($P_2$ and $P_3$). As a result, we have two partitions, each of which produces a join result of six records instead of 12, as in the hash-based partition algorithm. The range-based partition algorithm allows data to be partitioned into more than two partitions. With the "fragment–replicate" technique used, we can divide the workload of the partitions when generating the join results.

### 3.4. Multi-Dimensional Range Partition

Multi-dimensional range partition (MDRP) is an algorithm that combines a range partition and a random partition [6]. Mappers read input data from the datasets and create split vectors to divide the data into sub-ranges. The data will be put into a partitioning matrix corresponding to the sub-range values of the two datasets. Input data are distributed to the reducers based on this partitioning matrix, in which cells containing more data in the matrix will be subdivided and assigned to two or more reducers. The data flow of multi-dimensional range partition algorithm is presented in Figure 10.

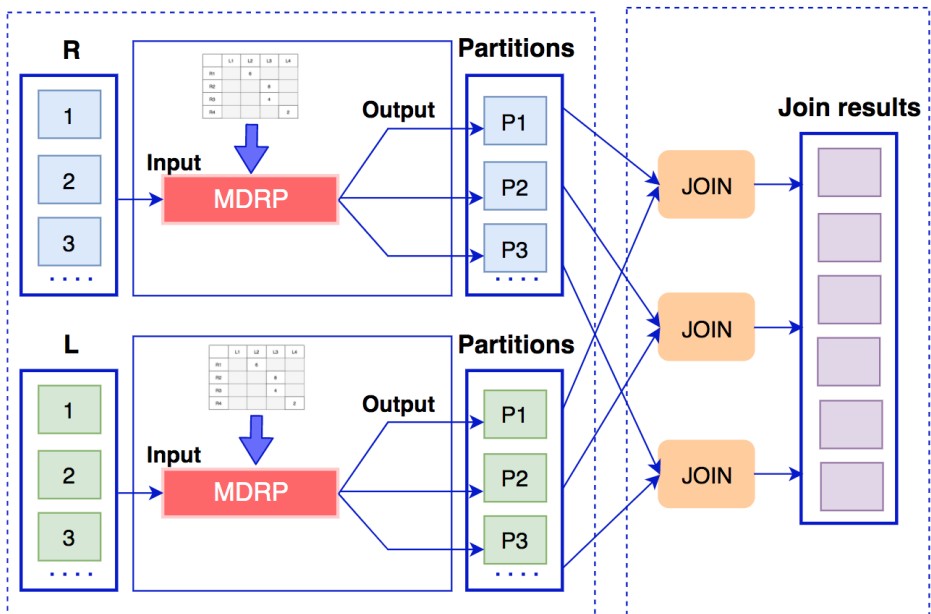

**Figure 10.** Data flow of multi-dimensional range partition algorithm.

### 3.4.1. Partitioning Matrix

The algorithm considers two datasets $R$ and $L$ with a join key of $k$. Suppose that there are $p$ partitions, and the input datasets will be divided into $p$ sub-ranges, i.e., $\{R_1, R_2, \ldots, R_p\}$ and $\{L_1, L_2, \ldots, L_p\}$. Creating sub-ranges of the two datasets is similar to creating sub-ranges with the range-based partition algorithm. For dataset $R$, the sub-range will include the entire domain of the join key $k$ from $\alpha$ to $\beta$, $\alpha < R.k < \beta$. Two special cases $R_1$ and $R_p$ will include sub-ranges $[-\infty, \alpha]$ and $(\beta, \infty]$, respectively.

As in the previous example, we have $R.k = \{1, 1, 2, 2, 2, 2, 4, 4\}$ and $L.jk = \{1, 1, 2, 2, 2, 3, 4, 4\}$ with $p = 4$ partitions. We get two split vectors $\{1, 2, 2\}$ and $\{1, 2, 3\}$ from the two datasets $R$ and $L$. In the partitioning matrix $M$, as shown in Figure 11, the $i$th row represents the sub-range $R_i$ and the $j$th column represents the sub-range $L_j$. The cell $(R_i, L_j)$ is classified into one of the following two groups: candidate and non-candidate. Candidates are cells that produce join results and non-candidates are cells that do not generate join results.

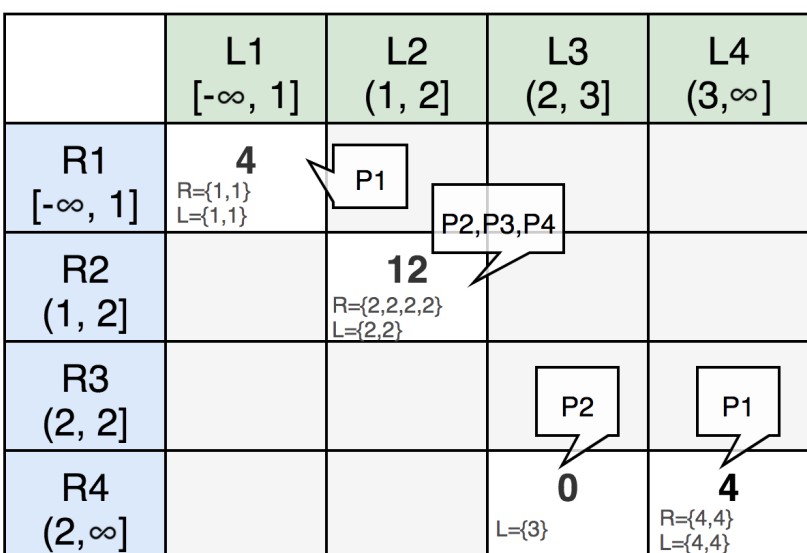

**Figure 11.** Mapping between cells and partitions.

$M(i, j)$ is the value representing the workload of a candidate cell $(R_i, L_j)$ to be processed. In the MDRP algorithm, $M(i, j) = |R_i \bowtie L_j|$, where $R_i$ and $L_j$ are records that belong to the same sub-range. In the partitioning matrix $M$, $M(2, 2) = 12$ because we have four records whose join key is 2 in dataset $R$ and three records with a join key of 2 in dataset $L$, which belong to the same sub-range $(R_2, L_2)$. Similarly, $M(1, 1) = 4$ because we have two records $\{r1, r2\}$ in dataset $R$ and two records $\{l1, l2\}$ in dataset $L$ that have join keys belonging to the same sub-range $(R_1, L_1)$. $M(4, 4) = 4$, since we have two records in dataset $R$ and two records in dataset $L$ that have join keys belonging to the same sub-range $(R_4, L_4)$. The partition matrix will help to ensure that the total number of cells $(R_i, L_j)$ and the total workload in each partition are relatively equal.

### 3.4.2. Identifying and Dividing Heavy Cells

Heavy cells in the partitioning matrix $M$ are cells $(R_i, L_j)$ that satisfy the conditions in Equation (1). That is, if the ratio of the workload of the cell $(R_i, L_j)$ to the total workload of the partitioning matrix $M$ is greater than or equal to $\frac{1}{p}$, this cell is called a heavy cell. The ratio $\frac{1}{p}$ is the optimal workload ratio for each partition. Therefore, if the partitioning matrix contains heavy cells, it will not be possible to balance the workload between the partitions.

$$\frac{M(i, j)}{\Sigma_{r=1}^{p}\Sigma_{l=1}^{p}M(r, l)} \geq \frac{1}{p} \tag{1}$$

As in the example, we have one heavy cell $(R_2, L_2)$. The total workload in the example is 20 and the optimal workload ratio is $\frac{1}{p} = \frac{1}{4} = 0.25$. The cell $(R_2, L_2)$ has workload $M(2, 2) = 12$; thus, it is a heavy cell ($12/20 = 0.6 \geq 0.25$). To ensure load balancing between the partitions, the heavy cell has to be divided into cells with smaller workloads. A quantity $\omega$ is defined as the optimal workload, as in Equation (2). After this value is defined, the heavy cells are split into $d$ cells, with the conditions shown in Equation (3).

$$\omega = \frac{\Sigma_{r=1}^{p}\Sigma_{l=1}^{p}M(r, l)}{p} \tag{2}$$

$$\frac{M(i, j)}{d} \leq \omega \tag{3}$$

We have $\omega = 20/4 = 5$. Therefore, the heavy cell $(R_2, L_2)$ will be split into three non-heavy cells, since ($12/3 \leq 5$). The heavy cells are split into several non-heavy cells that are partitioned into different partitions.

### 3.4.3. Partitioning the Non-Heavy Cells

- We have a list of non-heavy cells denoted by $C = \{c_1, c_2, \ldots, c_{|C|}\}$.
- Each $c_i \in C$ consists of a sub-range $(R_i, L_j)$, and its workload is denoted by $w(c_i)$. For example, a non-heavy cell $c_1$ is denoted by $(R_i, L_j, w(c_1))$.
- These non-heavy cells are partitioned into different partitions $P = \{P_1, P_2, \ldots, P_p\}$.
- A list of non-heavy cells assigned to partition $P_i$ is denoted as $C_i$.
- The number of non-heavy cells of $C_i$ is denoted as $|C_i|$.
- The total workload of partition $P_i$ is defined by $W_i = \Sigma_{c_l \in C_i} w(c_l)$.

We use the assign algorithm to distribute the non-heavy cells to the partitions. First, the non-heavy cells in $C$ are sorted in descending order of workload $w(c_i)$. For each non-heavy cell $c_i \in C$, we choose partition $P_i$ so that $P_i$ has the minimum number of non-heavy cells and minimum total workload. The algorithm is shown in Algorithm 4.

---

**Algorithm 4** Assignment algorithm

---

**Input:** $C = \{c_1, c_2, \ldots, c_{|C|}\}$, list of non-heavy cells sorted in descending order of $w(c)$
**Output:** $\{C_1, C_2, \ldots, C_p\}$, partitions of the non-heavy cells

1: **for each** cell $c_i \in C$ **do**
2:     $P_i$ = getNextPartition()
3:     $C_i = C_i \cup \{c_i\}$
4: **end for**
5: **return** $C_1, C_2, \ldots, C_p$

**getNextPartition**
**Output:** $P_{id}$, the partition selected for the non-heavy cells

1: minCell = $\infty$, minLoad = $\infty$, id = 0
2: **for each** partition $P_i$ **do**
3:     **if** $minCell > |C_i|$ **then**
4:         minCell = $|C_i|$
5:     **end if**
6: **end for**
7: **for each** partition $P_i$ **do**
8:     **if** $minCell = |C_i|$ and $minLoad < W_i$ **then**
9:         id = i
10:         minLoad = $W_i$
11:     **end if**
12: **end for**
13: **return** $P_{id}$

---

For example, we have a list of non-heavy cells sorted in descending order of workload, $C = \{(R_1, L_1, 4), (R_2, L_2, 4), (R_2, L_2, 4), (R_2, L_2, 4), (R_4, L_4, 4), (R_4, L_3, 0)\}$, and four partitions $P = \{P_0, P_1, P_2, P_3\}$. The three cells $(R_2, L_2, 4)$ are because the original cell is chopped into three cells. Initializing with $C_i = \{\}$ and $W_i = 0$, we have:

- $P_1 = \{(R_1, L_1, 4)\}, |C_1| = 1, W_1 = 4$
- $P_2 = \{(R_2, L_2, 4)\}, |C_2| = 1, W_2 = 4$
- $P_3 = \{(R_2, L_2, 4)\}, |C_3| = 1, W_3 = 4$
- $P_4 = \{(R_2, L_2, 4)\}, |C_4| = 1, W_4 = 4$
- $P_1 = \{(R_1, L_1, 4), (R_4, L_4, 4)\}, |C_1| = 2, W_1 = 8$
- $P_2 = \{(R_2, L_2, 4), (R_4, L_3, 0)\}, |C_2| = 2, W_2 = 4$

The result of assigning the non-heavy cells into partitions is shown in Figure 11.

### 3.4.4. Multi-Dimensional Range Partition

The MDRP algorithm is shown in Algorithm 5. The "fragment–replicate" technique is used for the join operation. Assume that a mapper receives a record $r3$ with the join key $r3.k = 2$ of dataset $R$; then, this record will belong to a cell $(R_2, L_2)$. However, this cell is divided into three partitions $P_2$, $P_3$, and $P_4$, since $(R_2, L_2, 4) \in C_2$, $(R_2, L_2, 4) \in C_3$, and $(R_2, L_2, 4) \in C_4$. So, we should use a fragment or a replicate for record $r3$. By counting the number of records in $R_2, L_2$ with join key $k = 2$, if the number of records of for dataset $R$ is greater than that for dataset $L$, then $r3$ will be fragmented (randomly selected partition), and the records in dataset $L$ will be replicated (replicated to both partitions).

---

**Algorithm 5** Multi-dimensional range partition algorithm

---

**Input:** an input tuple $r \in R$
**Input:** a partitioning matrix $M$
**Begin**

  1:  listCell = M.listCell(r.k)
  2:  **for each** cell c in listCell **do**
  3:      P = c.listPartition()
  4:      **if** M.fragment(r.k) == true **then**
  5:          i = random() % $|P|$
  6:          Output (P[i], r)
  7:      **else**
  8:          **for** i = 1 **to** $|P|$ **do**
  9:              Output (P[i], r)
 10:          **end for**
 11:      **end if**
 12:  **end for**
**End.**

---

## 4. Evaluation

### 4.1. Cost Model

Join computation cost is the total cost of several stages, including pre-processing, data reading, map processing, communication between nodes, reduction processing, and data storage. The parameters used in the cost model are shown in Table 1. The general cost model for the join computation of two datasets is described in Equation (4).

$$C(J) = C_{pre} + C_{read} + C_{map} + C_{tran} + C_{reduce} + C_{write} \tag{4}$$

where:

- $C_{pre} = 0$ (there is no preprocessing task of the three algorithms)
- $C_{read} = (|R| + |L|) \cdot c_r$
- $C_{map} = (|R| + |L|) \cdot c_m$
- $C_{tran} = |D| \cdot c_t$
- $C_{reduce} = MAX(C_{reduce(1)}, C_{reduce(2)}, \ldots, C_{reduce(n)})$
- $C_{write} = C_{write(1)} + C_{write(2)} + \cdots + C_{write(n)}$

  Some values ($|D|$, $C_{reduce}$, and $C_{write}$) are different in the three algorithms.

**Table 1.** Parameters used in the cost model.

| Parameters | Meaning |
|---|---|
| $n$ | Number of partitions |
| $c_r$ | Cost of reading/writing distributed data (units of time) |
| $c_m$ | Cost of processing map in memory |
| $c_d$ | Cost of processing reduction in memory |
| $c_t$ | Cost of communication between nodes |
| $\alpha_i$ | Ratio of skew-join key $i$ |
| $k$ | Number of skew-join keys |
| $|R|$ | Number of records in dataset $R$ |
| $|L|$ | Number of records in dataset $L$ |
| $|D|$ | Number of records in intermediate dataset $D$ |
| $|R'|$ | Number of records in $R$ after the mapping process |
| $|L'|$ | Number of records in $L$ after the mapping process |
| $|R''|$ | Number of skew records in $R$ after the mapping process |
| $|L''|$ | Number of skew records in $L$ after the mapping process |
| $|O|$ | Number of join result records in each partition |
| $|O'|$ | Number of join result records of $|R''| + |L''|$ |

**Table 1.** *Cont.*

| Parameters | Meaning |
|---|---|
| $C(J)$ | Total cost of join computation (units of time) |
| $C_{pre}$ | Cost of the preprocessing task |
| $C_{read}$ | Cost of data reading |
| $C_{map}$ | Cost of processing the map |
| $C_{tran}$ | Cost of communication between nodes |
| $C_{reduce}$ | Cost of processing the reduction |
| $C_{write}$ | Cost of data writing |
| $C_{reduce(i)}$ | Cost of processing in reducer $i$ |
| $C_{write(i)}$ | Cost of reading/writing data in reducer $i$ |

4.1.1. Hash-Based Partition

In this algorithm, records with the same key will come to the same reducer due to the hash function. Therefore, in this study, the skewed data are the records with a join key of 1, which will be processed by reducer 1.

- $|D| = |R| + |L|$
- $C_{reduce(1)} = (\frac{|R'|}{n} + \frac{|L'|}{n}).c_d + (|R''| + |L''|) \cdot c_d$
- $C_{reduce(2)} = \cdots = C_{reduce(n)} = (\frac{|R'|}{n} + \frac{|L'|}{n}) \cdot c_d$
- $\Rightarrow C_{reduce} = (\frac{|R'|}{n} + \frac{|L'|}{n}) \cdot c_d + (|R''| + |L''|) \cdot c_d$ (*)
- $\Rightarrow C_{write} = (\frac{|O|}{n} + |O'|).c_r + (n-1)(\frac{|O|}{n}) \cdot c_r$

4.1.2. Range-Based Partition

A range-based partition processes skewed data by randomly distributing skew records to the reducers and duplicating records with the same key in the remaining dataset.

- $|D| = |R| + |L| - \Sigma_{i=1}^{k}(|L| + \alpha_i) + \Sigma_{i=1}^{k}(|L| \cdot \alpha_i^2 \cdot n)$
- $C_{reduce(1)} = C_{reduce(2)} = \cdots = C_{reduce(n)}$
- $\Rightarrow C_{reduce} \approx (\frac{|R'| + |L'| - \Sigma_{i=1}^{k}(|L| + \alpha_i) + \Sigma_{i=1}^{k}(|L| \cdot \alpha_i^2 \cdot n)}{n}) \cdot c_d$ (**)
- $\Rightarrow C_{write} = n \cdot (\frac{|O|}{n}) \cdot c_r = |O| \cdot c_r$

The value of $\alpha_i \cdot n$ is equal to the number of segments with duplicate keys.

- If $\alpha = 0$, then $|D| = |R| + |L|$;
- If $\alpha = 10\%$ and $\alpha = 20\%$, the amount of intermediate data $|D|$ generated is negligible;
- If $\alpha = 50\%$, the intermediate data generated are significant, but the hash-based partition algorithm cannot be performed because the nodes processing the skewed data will be overloaded or even stop working. That is, in this case, a range-based partition is more efficient than a hash-based partition. Therefore, we only need to consider the cases of moderately skewed ratios and to compare the costs of the algorithms based on the performance of the reducers.

4.1.3. Multi-Dimensional Range Partition

A multi-dimensional range partition tends to divide data evenly, including the skew records in the datasets for the reducers.

- We compared the three algorithms based on the performance of the reducers as follows: $|D| \approx |R| + |L|$
- $C_{reduce(1)} = C_{reduce(2)} = \cdots = C_{reduce(n)}$
- $\Rightarrow C_{reduce} \approx (\frac{|R'| + |L'|}{n}) \cdot c_d$ (***)
- $C_{write} = n \cdot (\frac{|O|}{n}) \cdot c_r = |O| \cdot c_r$

### 4.1.4. Analysis

Since skew-join records occur often in reducer 1 and other skew keys are insignificant, reducer 1 determines the execution time of the algorithms. We compared the costs of join computation between the three algorithms.

- From (*) and (**),
  $\Rightarrow C_{reduce(HBP)} > C_{reduce(RBP)}$.
- From (*) and (***),
  $\Rightarrow C_{reduce(HBP)} > C_{reduce(MDRP)}$.
- From (**) and (***),
  $\Rightarrow C_{reduce(RBP)} > C_{reduce(MDRP)}$.

Therefore, we have Equation (5).

$$C(J)_{HBP} > C(J)_{RBP} > C(J)_{MDRP} \tag{5}$$

A range-based partition performs join operations with skewed data more efficiently than a hash-based partition thanks to the fragment–replicate mechanism. A multi-dimensional range partition is more efficient than a range-based partition thanks to the thorough skew processing of both datasets.

### 4.2. Experiments

#### 4.2.1. Cluster Description

We conducted experiments on a computer cluster with 14 nodes (one master and 13 slaves) at the Mobile Network and Big Data Laboratory of the College of Information and Communication Technology, Can Tho University. The configuration of each computer was with four Intel Core i5 3.2 GHz CPUs, 4 GB of RAM, 500 GB of HDD, and the Ubuntu operating system 14.04 LTS with 64 bits. The following versions of applications were used: Java 1.8, Hadoop 2.7.1, and Spark 2.0.

#### 4.2.2. Data Description

We generated experimental datasets with *scalar skew distribution*. The term "scalar skew" was introduced by Christopher Walton and his colleagues [19] when developing a taxonomy of skew effects. Scalar skew distribution was later used by other researchers for skew handling [6,23–25]. In this work, we generated six datasets, in which each had 100,000,000 records. The idea of scalar skew distribution is that, in 100,000,000 records, the skew-join key with a value of 1 appears in some fixed number of records. The remaining records contain randomly appearing join key values from 2 to 100,000,000. This will help us to easily understand which experiments are performed while keeping the output size constant over varying amounts of skew. The format of the datasets was plain text consisting of three fields separated by commas, the primary key, the join key, and other text (pk, jk, others). The six created datasets had different skew ratios for the experiments. The details of the datasets are presented in Table 2.

**Table 2.** Dataset description.

| Dataset | Number of Records | Number of Skewed Records | Skew Ratio |
|---|---|---|---|
| **DSkew100** | 100,000,000 | 100 | 0.000001 |
| **DSkew1K** | 100,000,000 | 1000 | 0.00001 |
| **DSkew10K** | 100,000,000 | 10,000 | 0.0001 |
| **DSkew100K** | 100,000,000 | 100,000 | 0.001 |
| **DSkew1M** | 100,000,000 | 1,000,000 | 0.01 |
| **DSkew10M** | 100,000,000 | 10,000,000 | 0.1 |

4.2.3. Evaluation Method

We used three algorithms—hash-based partition, range-based partition, and multi-dimensional range partition—in the three test cases. In each case of running the algorithms, we ran them three times to get the average execution time and evaluated the three algorithms based on their execution times. The join selectivity was the number of output records divided by the number of records in the cross product of the input relations. In this work, we kept the output size of $10^9$ for all three test cases.

- **Test 1** (high skew ratio): DSkew100 $\bowtie$ DSkew10M = 100 * 10,000,000 = $10^9$
- **Test 2** (average skew ratio): DSkew1K $\bowtie$ DSkew1M = 1000 * 1,000,000 = $10^9$
- **Test 3** (low skew ratio): DSkew10K $\bowtie$ DSkew100K = 10,000 * 100,000 = $10^9$

$$JoinSelectivity = \frac{|(R \bowtie_c L)|}{|(RxL)|} = \frac{|(R \bowtie_c L)|}{(|R| * |L|)} = \frac{10^9}{10^{16}} \qquad (6)$$

4.2.4. Analysis of the Results

We examined the performance of the algorithms in the join tests with different ratios of skewed join keys. The differences in the execution times of the join tests are shown in Figure 12. In the first join test with a high skew ratio (Test 1), the multi-dimensional range partition performed better than the others. The hash-based partition algorithm was the worst, as it was 1.59 times slower than the multi-dimensional range partition and 1.44 times slower than the range-based partition. In the second join test (Test 2), the multi-dimensional range partition was slightly better than the other two algorithms. Nevertheless, the performance of the three algorithms was almost equivalent in the case of the average skew ratio. In the last join test with a low skew ratio (Test 3), the hash-based partition gave a better performance than the others. The multi-dimensional range partition was the worst in this case, as it was 1.7 times slower than the hash-based partition. The range-based partition appeared to have an average performance in comparison with the other two algorithms in the three test cases. Regarding Formula (5), the experimental results were appropriate for the cost models presented above.

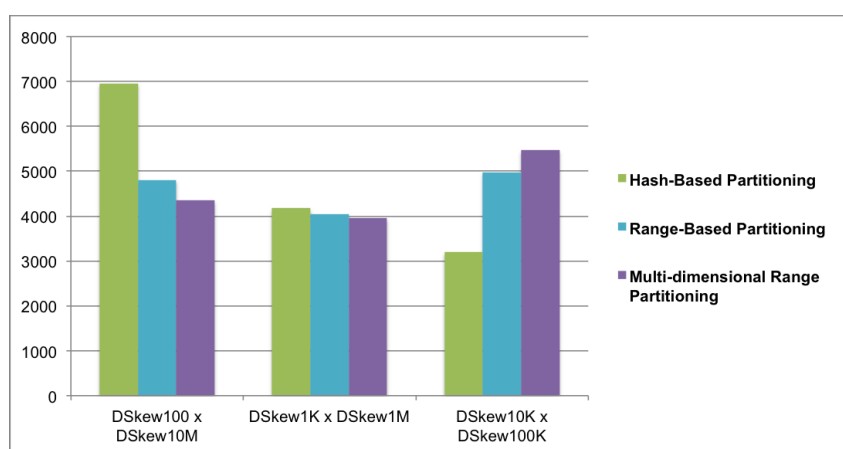

**Figure 12.** Execution time of the three algorithms (seconds).

Considering the advantages and disadvantages of the algorithms, with the hash-based partition, the advantage of this algorithm is that it is easy to implement and commonly used in join computation. However, it is not ideal for handling join operations with a high skew ratio. Tuples with the same join keys are hashed to the same reducers, leading to seriously imbalanced tasks. As can be seen, datasets with more skew-join keys had a worse performance; on the contrary, datasets with fewer skew-join keys had a better performance.

In the case of the range-based partition, the tuples with more skew data were selected for range determination. It is quite easy to determine sub-ranges with a small cost with this

method; thus, it has been widely used to deal with data-skew problems. This method is more efficient than the hash-based partition algorithm, but it is not a good choice in the case of datasets with a low skew ratio. A limitation of this algorithm is that the determination of the sub-ranges does not take into account the size of the join results; thus, a joining product skew can arise.

The multi-dimensional range partition overcomes the limitation of the range-based partition by using a partitioning matrix instead of a split vector. The cross products of the sub-ranges represented by cells in the partitioning matrix can be estimated to avoid the oversizing of the join results. With this algorithm, the greater skew ratio of the datasets, the more the performance is improved. Conversely, datasets with fewer skew-join keys incur a higher cost of creating a partitioning matrix, recalculating heavy cells in the partition matrix, and assigning cells to reducers.

## 5. Conclusions

Implementations of MapReduce are used to perform many operations on very large datasets, including join operations. It has become a prominent parallel and distributed programming model for efficiently handling massive datasets. The major obstacle of the model for join processing is data skew. If the data are significantly skewed, with some common join keys appearing to have a greater frequency than the others, the reduction tasks of these join keys will finish later than those of the others. Thus, any benefits from parallelism become meaningless. There are some algorithms that have been proposed to solve the problem of skew joins. Several surveys on solutions for skew joins have been made, but an adequate and systematic comparison in the Spark environment has still not been presented. Thus, this work was designed to provide a comprehensive comparison of several skew-join algorithms with mathematical models and experiments. We fully evaluated the hash-based partition, range-based partition, and multi-dimensional range partition algorithms in MapReduce on the Spark framework. We provided an analysis of the advantages and disadvantages of each algorithm. The cost model built was an important theoretical basis for the evaluation and comparison of the skew-join algorithms. Lastly, the experiments were conducted in Spark, the new generation of Big Data processing. Through the cost models and experimental results, this research presented a comparison of the three algorithms. This is a highly scientific contribution, since join operations are commonly used in Big Data environments. In the scope of this work, we only provided an evaluation of the three algorithms. It is necessary to conduct an investigation and evaluation of more skew-join algorithms to have an overview of the problem of skewed data processing. In addition, Apache Spark 3 was introduced to dynamically handle skew in sort–merge join operations by splitting and replicating skewed partitions. It would be interesting to compare the data-skew handling provided by Spark and the user-defined data-skew handling with other algorithms.

**Author Contributions:** Conceptualization, T.-C.P. and A.-C.P.; methodology, A.-C.P., H.-P.C. and T.-C.P.; software, T.-C.P. and T.-N.T.; validation, A.-C.P. and H.-P.C.; formal analysis, A.-C.P. and T.-C.P.; investigation, A.-C.P. and T.-C.P.; resources, A.-C.P. and T.-C.P.; data curation, T.-N.T. and H.-P.C.; writing—original draft preparation, T.-C.P. and T.-N.T.; writing—review and editing, H.-P.C. and A.-C.P.; visualization, T.-N.T. and H.-P.C.; supervision, T.-C.P.; project administration, A.-C.P. All authors have read and agreed to the published version of the manuscript.

**Funding:** This research received no external funding.

**Institutional Review Board Statement:** Not applicable.

**Informed Consent Statement:** Not applicable.

**Data Availability Statement:** Data are available on request by contacting the corresponding author.

**Conflicts of Interest:** The authors declare no conflict of interest.

## Abbreviations

The following abbreviations are used in this manuscript:

RDD     Resilient Distributed Dataset
HDFS    Hadoop Distributed File System
HBP     Hash-Based Partition
RBP     Range-Based Partition
MDRP    Multi-Dimensional Range Partition

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
