# Peer review of "Comparative Analysis of Skew-Join Strategies for Large-Scale Datasets with MapReduce and Spark"

_applsci, doi:10.3390/app12136554_

Round 1

Reviewer 1 Report

This paper studied 3 data partitioning algorithms , established the cost model for each and compared the performance on Spark. The strategy of this study is scientific and the presentation is neat and clear. My comments are minor and should be addressed before publishing. Please find the following:

-The font in Fig 4 should be increased.

-It would be better to bring up the Abbreviations to the beginning of the paper.

-The authors are suggested to put more weights on analyzing the simulation results, for instance, Multi-dimensional range partition is better than the other 2 algorithms, but by how much? The part of results analysis is usually more insightful and in depth than what showed in this paper. 
-Besides these 3 algorithms, there are also other algorithms for skew data join operations, justifications for excluding them in this study need to be provided and highlighted.

Reviewer 2 Report

The presented article is very clearly written. Well-constructed graphs facilitate the reception of the difficult analytical issue presented by the authors. However, I have doubts about presentation of the selected issue, because in its current form it resembles the content of the academic textbook and not the empirical research text.

I also believe that the goal set by researchers indicating the assessment of the usability of the described methods has been briefly realized. The ending and conclusions of the article also require development.

Reviewer 3 Report

Data skew is a serious problem not only for joins in Spark programs. Joins of data sets highly skewed on the join key result in long running times of the executor processing the partition containing much more data than the others. This might ultimately lead to timeouts or even crashes. Thus, the skewed data must be handled in order to achieve a good performance of a user's program.

In this paper the authors compare three partitioning algorithms to redistribute a skewed data set so that the performance of a join is improved. The authors compare hash partitioning, range partitioning, and multi-dimensional partitioning that were introduced years ago for DBMSs.

Strong points:
- Spark is widely used for Big Data processing and skewed joins are a real problem
- The paper compares different partitioning strategies to handle different skew ratios
- the authors develop a cost model that could help to decide how to partition the data

Weak points:
- I miss a discussion of the evaluation results: The evaluation was performed on differently skewed data sets and showed no clear winner. For heavily skewed data, MDRP performed best, hash-based worst. For a low skew ratio MDRP was worst while hash-based partitioning was best. How can a user decide which one to choose?

- The authors did not use or even evaluate their cost model in the evaluation. It would be useful to see if the model can be used to decide which partitioning algorithm to apply for a given data set.

- The authors use Spark 2, whereas Spark 3 was released two years ago and has some built-in support for skewed joins. In my opinion, a comparison if the used partitioning approaches improve the native Spark 3 engine should be performed.

- In general, a comparison to other approaches is missing in the evaluation. As the referenced algorithms show, skew handling has been a problem that has already been studied for parallel databases.

Some minor points:
- line 268: I think  |R| / p-1 should be in a \lfloor , \rfloor
- Eq(1): boths sums use "i", should be l and r
- Line 367: the "2" is outside the parentheses
- Code listings have a "Table" caption

Round 2

Reviewer 2 Report

i don't have any further comments. thank you for the article correction

Author Response

Dear Reviewer,

Thank you very much for your precious time and consideration on our manuscript titled: “Comparative Analysis of Skew Join Strategies for Large-Scale Datasets with MapReduce and Spark” (ID: applsci-1739128). We again appreciate your kindness in helping us improve the manuscript.

Yours sincerely,

Thuong-Cang PHAN

Yours sincerely,

Thuong-Cang PHAN

Reviewer 3 Report

Thank you for the response, clarification, and update of the paper.

The paper compares three partitioning strategies with the goal to optimize a join operation in Spark. I think that this is interesting for the readers.

However, I my opinion, a comparison with Spark 3 (without changing Spark) and evaluation of the cost model would further improve the paper.

Author Response

Dear Reviewer,

Thank you very much for your precious time and consideration on our manuscript titled: “Comparative Analysis of Skew Join Strategies for Large-Scale Datasets with MapReduce and Spark” (ID: applsci-1739128).

We agree with your constructive comments. Currently, we run experiments on a Spark cluster of 14 nodes at the Mobile Network and Big Data Laboratory of the College of Information and Communication Technology, Can Tho University. The cluster used Spark version 2 is shared by several research teams. The change to Spark version 3 requires the consensus of other research groups. This is an obstacle for us right now thus we will conduct experiments on Spark 3 in the near future. We have noted this in the conclusion of the article.

We again appreciate your kindness in helping us improve the manuscript.

Yours sincerely,

Thuong-Cang PHAN
